# Development of AI-Augmented optimization technique for analysis & prediction of modal mix in road transportation

**Huma Rauf**[1,2]*, **Muhammad Umer**[3]

**1** Department of Business & Engineering Management, Sir Syed CASE Institute of Technology (SS-CASE-IT), Islamabad, Pakistan, **2** Department of Student Affairs, Rawalpindi Women University, Rawalpindi, Pakistan, **3** Faculty of Business & Engineering Management, Sir Syed CASE Institute of Technology (SS-CASE-IT), Islamabad, Pakistan

* humafawad6@gmail.com

**Data Availability Statement:** All relevant data are within the manuscript and its Supporting Information files.

**Funding:** The author(s) received no specific funding for this work.

## Abstract

Transport sector contribution to global emissions is a known fact, however, the mitigation path to achieve nationally determined goals for carbon reduction is often not specified, A simplified technique based on minimax optimization using Grey relational grade and Random forest narrows down on most contributing input variables from twelve road transport modes. This is a region-specific, scenario-based technique applied to north Punjab, Province of Pakistan that first categorizes modes based on their emission and then integrates with AI modeling using Deep Neural Network to develop sustainable trade-offs for carbon reduction. The output parameter translates the problem into a systematic iterative technique that predicts optimization options with different scenarios to bring out an environment-friendly transport mix. A 25% reduction applied to the five most emission-releasing modes like Diesel Light and Heavy Duty vehicles, Gas Light and heavy-duty vehicles, and Gas-Cars results in 16.54 MT of Carbon dioxide which is 54.35% reduced to the predicted 36.24 MT for the year 2044. Similarly in another scenario replacing 25% Gas and Diesel Light Duty vehicles respectively by adding 50% Petrol Light Duty vehicles leads to 18.94 MT of emissions which brings the emission value in 2044 at par with emission releases of the year 2014. The technique offers a forward path that allows environment-friendly modal mix combinations based on business-as-usual to offer transport mix solutions for carbon reduction. It is a generalized model that is based on a customized transport mix. Future studies can also be applied to intermodal tradeoffs like rail, air, waterways, etc.

## 1. Introduction

Fossil fuel drives the modern-day world [1] producing emissions that are way beyond the natural cycle of absorption and renewal, disturbing the biological balance of life on the planet. Environmental sustainability is a genuine concern worldwide and countries are reaching out for sustainable technologies and solutions [2]. This research focuses on the transport sector which contributes 25% of the global share of emissions [3] and is growing exponentially as

**Competing interests:** The authors have declared that no competing interests exist.

releases are 71% greater than the emissions released in the year 1990s [4]. The current emission calculation models do offer comprehensive solutions but very few countries are calculating the impact in actuality, resulting in a research gap for models that offer simple to-calculate emission solutions. The novelty of this research is its direct approach to narrowing down the most damaging input variable and applying optimizing techniques for quick-fix solutions. The region of this study is the north Punjab region of Pakistan in South Asia which has an average recorded warming of 0.75˚C and is considered one of the most vulnerable areas hit by environmental deterioration globally. The transport sector of Pakistan is accounted for 29% of the total $CO_2$ releases of the country [5] with road transport catering to 94% of the overall passenger transport and 98% of the overall freight transportation while the remaining 6% of passenger and 2% of freight requirement is being met by rail and air modes [6]. Optimizing the modal mix for this area would allow significant carbon reduction targets to be met on a long-term basis for the country.

The current global energy demand derived from fossil fuels is expected to rise by 70% for industries, 29% for commercial buildings, and 20% for the transport sector [7]. The transport sector is heavily dependent on fossil fuels as 93% of energy comes from oil and global energy consumption rose from 23% in 1971 to 29% in 2017 [8] resulting in 80–90% of emissions coming from road transport, 5–8% from rail, 1–2% from air traffic, and 1% from water transport. United Nations Framework Convention on Climate Change (UNFCC) during its convention in Paris in the year 2015 gave Nationally Determined Contribution (NDC) goals to 195 signatory countries to work on reduction targets [9]. This would contribute to the overall global targets for reduction in greenhouse gases (GHG) limiting temperature rise below 2˚C by the end of this century [10] (S2 Fig, CDKN, 2016). The combustion process releases long-lived and short-lived anthropogenic influencers that are causing climate deterioration [11], ozone layer depletion, smog in cities, rise in global temperatures [12], contamination of water and air sources [13], health problems, etc. However, contrary to the targets defined, global economies are challenged with rising income levels, increased urban settlements, and enhanced activities fueling the rise in demand for energy that is projected to increase by more than a quarter by the year 2040 [7].

The emissions are comprised of Carbon dioxide (CO2), Nitrous Oxide (NOX), Sulphur dioxide (SO2), Volatile Organic Compound (VOC), Particulate Matter (PM2.5 & PM10), Methane (CH4), and Ammonia (NH3) in different proportions [14] with every pollutant causing different impacts and CO2 or GHG is accounted for its warming potential. Air pollution is a transboundary phenomenon and pollutants irrespective of their place of origin become part of the troposphere and bring impacts like ozone layer depletion, warming, acidification, eutrophication, smog, etc. Road transport emissions releases are at low altitudes with a relatively low degree of dispersion resulting in concentrated hotspots, particularly in urban areas, besides the vehicles are not stationary so it becomes extremely difficult to combat the damages. The improvement of fuel and technology has achieved a two-fold reduction in emissions during the past two decades yet the usage of transport both for commute and freight levels has increased many folds [15].

Pakistan, as Berkley Earth data suggests would face significant environmental impacts due to non-uniform warming patterns as western areas experience 1.3˚C of warming compared to 0.9˚C in the southern part [16]. The overall energy demand of 8.70 Mtoe [106.7 TWh] is increasing at an annual growth rate of 6.60% would be 24.19 Mtoe [297.2 TWh] by the year 2050. The transport sector in Pakistan is an integral part of the country's economy as it contributes to 10% of the GDP besides creating 6% of employment opportunities. Besides the fuel, the reduction targets can be achieved through the efficient road network, traffic management, driving pattern modification, monitoring hot spot formations and relevant policy

formulations, etc. [17, 18]. Another mechanism that can be adopted for reduced emissions is the choice of mode, modal mix, and intermodal combination between land, sea, and air modes, and is an applicable practice by different countries [19]. This intermodal shift too has its tradeoffs and each change would bring an impact on overall emissions released like 1% increase in air passengers would account for 0.21% increased emissions while the same increase in the rail sector would increase emissions by 0.32% [8].

This research focuses on the role of the transport sector's contribution to the Nationally Determined Contribution (NDC) of a country as, even though 81% of 195 signatories of the Paris Convention agree that transport emissions are a major contributor to air pollutants, only 10% of these countries have submitted specific mitigation plans for incorporation in their national action plans [10]. The objective of the research is to suggest a simplistic model that acts as a carbon ceiling monitor on the current modal mix and suggest ways to bring reduction based on the business-as-usual basis to strategically plan for the modal mix combinations that can achieve reduction without relying on any fuel or technological transformation. The research questions are broadly classified as below:

- Can one significant variable be specified as a predictor variable for GHG emissions?

- Would a mini-max optimization model be combined with AI techniques to generate future modulation of emissions?

- Can this model be used to achieve reduction targets as per specified in the nationally determined contribution?

This is a region-specific modeling technique and can be applied by researchers and policy planners for the best combination of modal mix to meet desired INDC goals assigned by UNFCC. The baseline scenario of transport emissions is taken from the year 2015, which when projected to the year 2030 calls for 1400 MT of $CO_2$ to be reduced from the atmosphere while a two-degree scenario (2DS) target would require a further reduction of $CO_2$ by 600 MT.

## 2. Literature review

The transport sector plays a pivotal role in developing cross-cutting scenarios for building strategies to achieve de-carbonization and carbon neutrality [20]. Supply chain tools revolve around trade-offs between cost competitiveness, efficient delivery time, and efficient routes. To visualize these modalities, researchers built multimodal scenarios that can assist in the performance variability analysis for better decision-making [21]. The focus of their research is on the development of AI Integrated optimization tools that monitor emission reduction besides cost and time [22]. Most of this research end up in policy imposition statement that calls for behavioral modification through penalties like the carbon tax, ban on internal flights, etc, which would yield far better results if focused more on the control applied on the most proliferating modes of transport like a "car repression" strategy [23]. Demand shifts and behavioral modifications are hard to achieve but we do observe these pattern changes when a calamity occurs like during COVID-19, consumer demand patterns shifted to online buying [24], and that posed a challenge for fast-moving consumer goods companies to come forward with an appropriate distribution channel for smooth and efficient delivery of these goods [25]. Most of the logistics during Covid revolved around the supply chains resulting in comparatively lesser levels of pollutants and aerosols which also proved to be beneficial in reducing the spread of viruses as researchers found a direct correlation between PM2.5 and COVID-19 [26]. Similar strategies to apply controls on emission suggest policies; a combination of unimodal and intermodal transport [27]; containerization policy for freight [28], truck weight regulation,

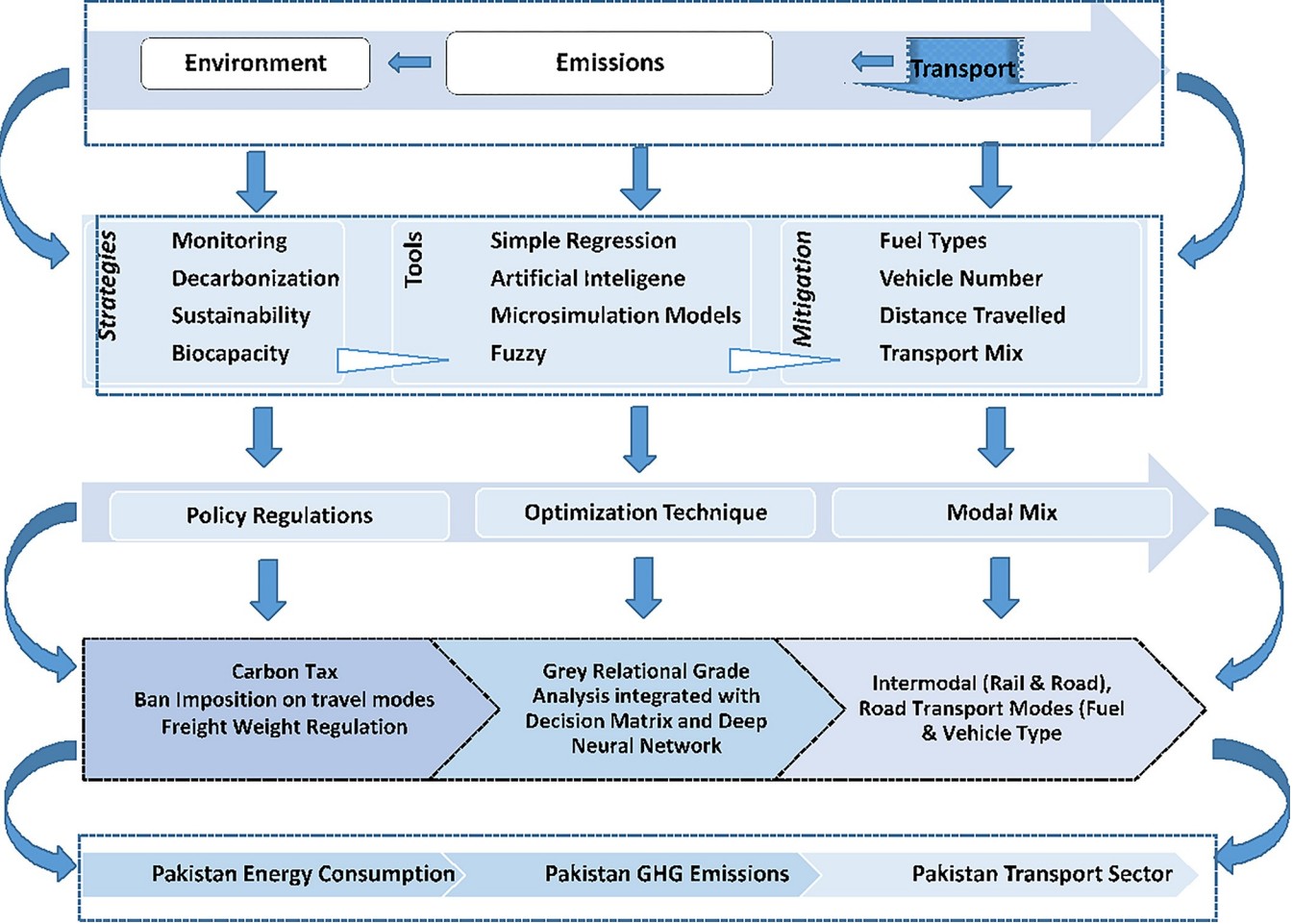

**Fig 1. Overall literature review.**

overload ratio, etc. [29]. Besides freight, commute demand has also experienced a surge due to dispersed activity-based lifestyles, shifting employment patterns, and changing family demographics [30] so a similar multi-modal tradeoff can also be applied to public transport service, as well to achieve carbon reduction targets [23].

Economic and environmental policies go hand in hand and trickle down to institutional-level performances where organizations contribute towards sustainable production of quality products in a country to boost exports and to play their role in the economic progress of their country [31, 32]. This calls for sustainable practices to be implemented in production, however, the bio capacity of any region is heterogeneous so homogeneous policies to sustain, support and regenerate cannot be implemented [33]. Emissions originating from localized hubs or scattered platforms [34] all add to a common domain called the atmosphere. A graphical representation of the literature review is shown in Fig 1 below:

Various techniques have been developed over the last century for emission calculation ranging from simple regression [35] to least square support vector machine [36], Artificial neural network (ANN) [37] to fuzzy logic methods [38], etc, The application of the technique depends on the spatial and temporal requirements for which a range of micro and macro simulation models can be used [39]. Microsimulation model's emission calculation have been standardized in different

parts of the world; in Europe "Computer Program to calculate emissions from road transport" (COPERT) and "Handbook of Emission Factors of Road Transport"(HBEFA) are mostly used [40] while in the US, an EPA developed technique "Motor Vehicle Emission Simulator" (MOVES) [41], is mostly used. When these micro-simulation models incorporate multiple variables from different spheres like social, technological, economic, environmental, traffic management and road designs, etc., the model becomes complex like system dynamic models [42], techno-economic models, and integrated assessment models [39], etc. To draw future projections on emissions, it is common to use multilinear regression (MLR) and multiple polynomial regression (MPR) based on business as usual (BAU), and projections are subject to assigned goals or policy that may be verified through these techniques [35]. These models connect emission models with transport, environment, and other integrated overlapping models [43].

Grey relational optimization tool has its origin in Grey System theory in the year 1982 and is applied to systems having incomplete or undetermined information. The usage of grey relations, grey elements, or numbers is a typical feature that relates to grey uncertainty usage. It turns the disorderly raw data into regular series data that can easily replace stochastic processes to find real-time techniques for prediction, decision-making, relational connections, and industrial and multi-dimensional applications [44]. Grey Relational Analysis (GRA) is u for generating qualitative and quantitative relationships among complex factors which often has insufficient information and generates a single response termed Grey Relational Grade (GRG) to develop an optimum combination of input and output for multi-objective problems. GRG serves as a reference grade that represents the relative distance between different variables and can ascertain the comparative influence of multiple factors on the output [45]. It can be used combined with other techniques like the Taguchi method to generate Taguchi Grey-relational analysis [46] where they applied it to optimize the design and operational parameters of an engine [47]. In China, GRA analysis is widely used in establishing the relationship between transportation with energy consumption and $CO_2$ emissions data from multiple provinces [45]. GRA Grey Theory is used widely in analytics in multiple hybrid modes like the novel Partial Least Square Model combines with Grey and Markov theories for PLS-Grey-Markov Models [48]. Another similar combination is achieved by combining GRA with principal component analysis (PCA) and long-short memory (LSTM) to evaluate $CO_2$ emissions [49]. Likewise, neural networks are modern statistical tools that bring forward optimized solutions by effectively handling the non-linear behavior of inputs and output variables. Our study focuses on the transport sector and $CO_2$ which is the prime emission impacting global warming and then nitrous oxide is also a significant pollutant released and the projections can be seen S3A and S4B Figs. Different sectors have different prime pollutant releases as $NO_x$ is released most from the transport sector; $SO_2$, VOC, and PM mostly originate from the energy sector. Narrowing it down to the transport sector emissions vary with the choice of fuel; NOx is the major release from diesel, VOC is released more from Petrol, $SO_2$ depends on the fossil fuel grades, PM or burnt carbon in different micron sizes is a result of unburned fuel particles, wear and tear of tires and brakes, etc.

## 3. Materials and methods

The methodology sequence is depicted in Fig 2 below:

### a. Identification of input and output variables

The overall data preparation for Input variables is shown below in Fig 3:

$$E_{i,p} = \sum_k \sum_m A_{i,k}\, ef_{i,k,m,p} x_{i,k,m,p}$$

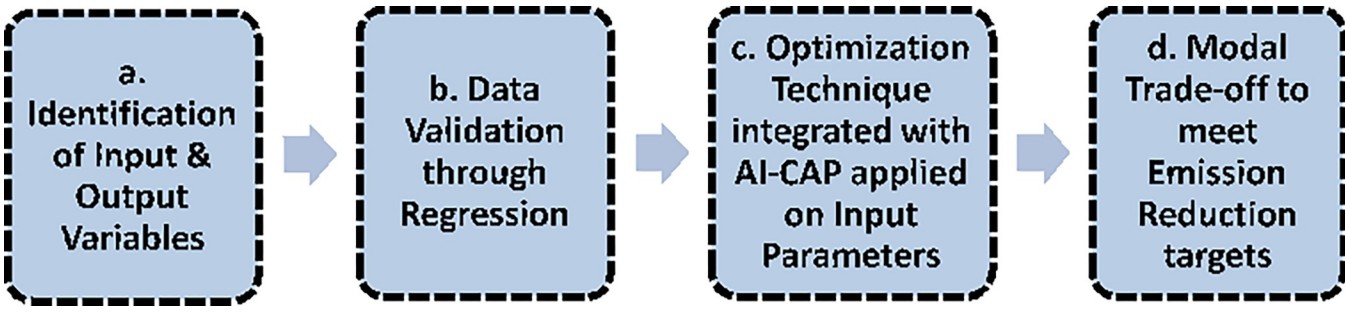

**Fig 2. Methodology of research.**

$ef_{i,k,m,p}$ Emission Factor for pollutant $p$ in country $i$ with measures m for activity $k$

$A_{i,k}$ Activity $k$ in country $i$.

$x_{i,k,m,p}$ Activity $k$ share in country $i$, with control measures, m for pollutant $p$

$E_{i,p}$ Emission relating to pollutant p and country $i$

Petrol Two-wheelers Motorcycles & Mopeds (P-MP, P-MC),

Petrol, Diesel & Gas Light-Duty Vehicles & Cars (P-C, P-LD, G-C, G-LD, D-C, D-LD)

Gas & Diesel Heavy-Duty Vehicles & Bus (G-B, G-HD, D-B, D-HD)

Energy consumption and distance traveled are comparable units for all vehicle types but for bringing homogeneity in comparing multiple transport modes [50], a widely used conversion for equivalence called passenger car unit (PCU) is applied. Though PCU also varies for static and dynamic situations, road design, driving conditions, lanes, different regions of the world, etc [51], we followed Singaporean conversion as in Table 1:

## b. Regression analysis

Statistical Analysis generated a multivariable regression model from the cumulative impact of three input variables on carbon emission as singular output and the R square value obtained is 0.9997 with normal probability data as a straight diagonal line confirming normally distributed data as seen in Fig 4:

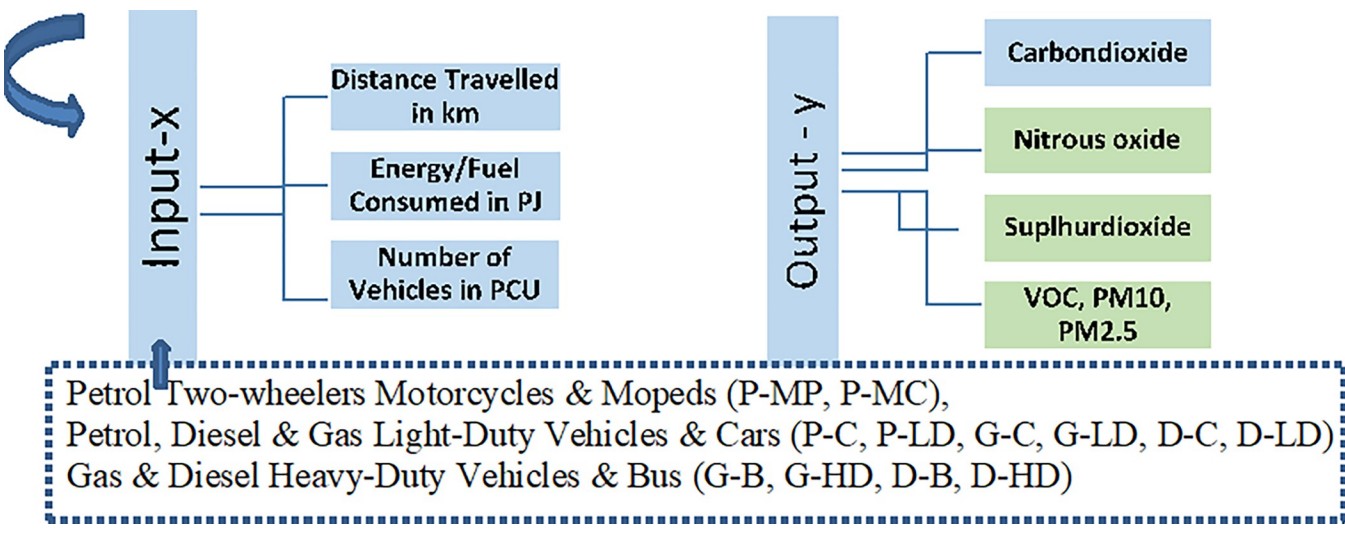

**Fig 3. Data preparation for input output variables.**

**Table 1. PCU equivalent for multiple transport modes.**

| PCU (Passenger Car Equivalent/Unit) | Singapore | UK |
|---|---|---|
| Motorcycles | 0.65 | 0.75 |
| Passenger Cars | 1.00 | 1.0 |
| Light Goods Vehicle | 1.53 | 1.0–2.0 |
| Heavy Goods Vehicle | 2.75 | 2.0 |
| Buses/Coaches | 2.75 | 3.0 |

### c. Transport optimization integrated with AI Augmented Climate Change Analysis & Prediction technique (AI-CAP)

The whole process is depicted in the flow chart below in Fig 5:

### d. Tradeoff

Few modes that are high on emissions due to their fuel type or vehicle type are required to fade out in the future so these may be replaced by modes that cause the least damage as depicted in Fig 6 below. These tradeoffs would generate targets to be set for meeting the INDC Goals of the UN.

Transport sector emissions are growing at an average rate of 1.7% from the year 1990 to 2021 and to reach a targeted net zero emissions by 2050, a 3% reduction in emissions has to be achieved every year (IEA, 2022). The research is designed to achieve a simple model that only considers one variable and uses a solution in hand based on business as usual to put a check on the modal use of vehicles on the road instead of waiting for alternate fuel or technology solutions to help us achieve sustainability. Society would see gradual but long-term benefits, particularly in terms of public health. It would reduce the social cost factor by providing health benefits to people suffering due to pollution and related health disorders like respiratory tract diseases, skin ailments, allergens, and carcinogens. With the slowing down of the climate

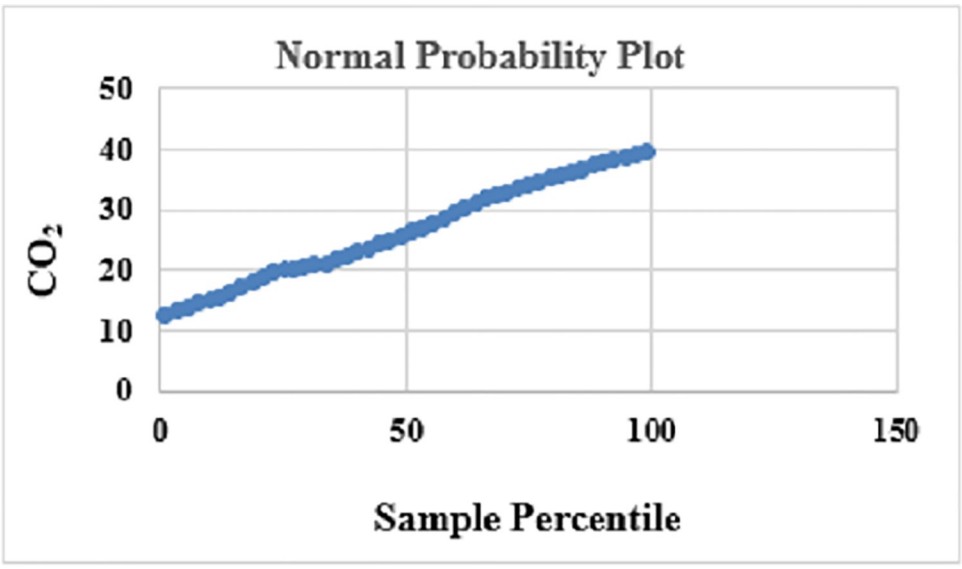

**Fig 4. Normal probability plot analyzing $CO_2$ emissions with input variables fuel consumed, vehicle number, and distance traveled.**

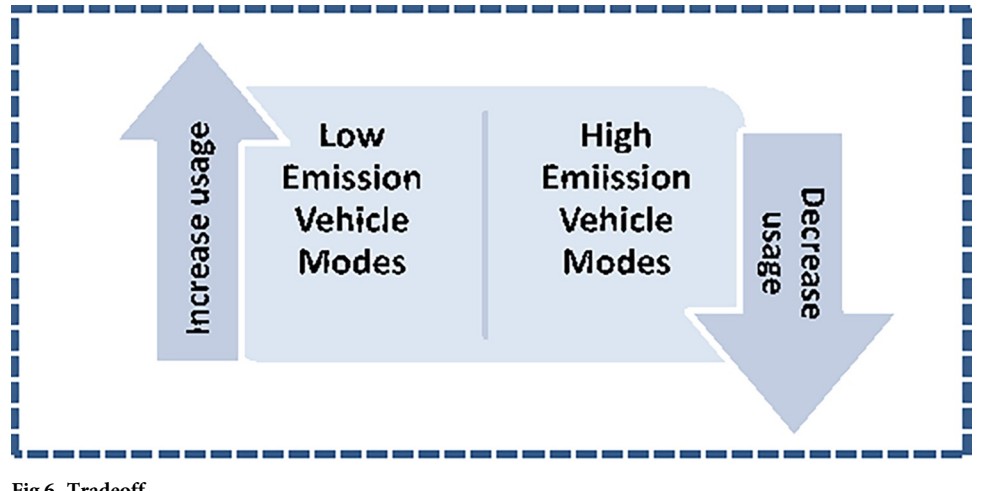

**Fig 5. Optimization sequence of GRG, random forest, and deep neural network.**

**Fig 6. Tradeoff.**

change clock, the planet's biological ecosystem would start healing gradually towards sustainability and the impact would be seen in the form of decreasing rate of glacial melting & rising sea levels, fewer episodes of extreme weather disasters like floods, hurricanes, and drought, etc. This restoration would also be seen in the restoration of seasonal growth of crops, flower blooming season, bird migration pattern, and restoration of marine and aquatic plant lives.

## 4. Results & discussion

Statistical regression on the combined input variables with $CO_2$ emission as the output variable generated a model with an $R^2$ value of 0.9997 and a p-value less than the alpha value as below:

$$y = 0.066 \text{ Xf} + 0.000145 \text{ Xn} - 0.02208 \text{ Xd} + 1.3344099$$

y = $CO_2$ emissions in MT/year
Xf = Fuel Consumed in PJ/year
Xn = Passenger Car Unit in 1000s/year
Xd = Distance Covered in $10^9$ km/year

The regression relationship however does not characterize an optimized combination for minimal carbon emissions, therefore, Grey relational analysis is carried out initially on the three input variables combined data set based on PCUs to analyze the first applied on the combined data set values based on PCUs. Here, the number of km traveled is taken as higher the better as maximizing function and calculated as follows to normalize the data;

$$x_i^*(k) = \frac{x_i^0(k) - \min x_i^0(k)}{\max x_i^0(k) - \min x_i^0(k)}$$

Similarly, both the number of vehicles in PCUs and Energy consumption is taken as the lower the better with the application of a minimizing function while normalizing the data as below:

$$x_i^*(k) = \frac{\max x_i^0(k) - x_i^0(k)}{\max x_i^0(k) - \min x_i^0(k)}$$

Grey relational coefficient ($\xi$ is taken as 0.5)

$$\xi_i(k) = \frac{(\Delta \min + \xi . \Delta \max)}{(\Delta_{oi}(k) + \xi . \Delta \max)}$$

Grey relational grade ($\omega_k$ is taken as 1 normally)

$$\gamma_i = \frac{1}{n} \sum_{k=1}^{n} \omega_k(k) \xi(k)$$

The optimization technique is applied with a multi-objective approach for lowering $CO_2$ emissions. Grey relational analysis carries out multi-objective optimization following three steps first normalizing the data then establishing Grey relational grade (GRG) and ranking the dataset based on the best outcomes. This would establish the best multi-objective prediction and optimization sequence and would develop a multiple regression model for GRG besides predicting the impact of the most significant contributor to this study.

The calculation of GRG is done as below in Table 2:

**Table 2. Grey relational analysis for three input variables.**

| Year | Data set | | | Normalizing | | | Deviation Sequence | | | Grey Relational Coefficient | | | Grey Relational Grade | |
|---|---|---|---|---|---|---|---|---|---|---|---|---|---|---|
| | FUEL PJ | PCU 000 | KM x 10$^9$ | FUEL PJ | PCU 000 | KM x 10$^9$ | FUEL PJ | PCU 000 | KM x 10$^9$ | FUEL PJ | PCU 000 | KM x 10$^9$ | GRG | RANK |
| 2005 | 181.2 | 3835.9 | 45.24 | 1 | 1 | 0 | 0 | 0 | 1 | 1 | 1 | 0.333 | 0.778 | 1 |
| 2006 | 191.5 | 4195.7 | 53.36 | 0.977 | 0.945 | 0.051 | 0.023 | 0.055 | 0.9494 | 0.9557 | 0.9 | 0.345 | 0.734 | 2 |
| 2007 | 201.7 | 4555.6 | 61.48 | 0.954 | 0.89 | 0.101 | 0.046 | 0.11 | 0.8989 | 0.9152 | 0.82 | 0.357 | 0.697 | 3 |
| 2008 | 211.9 | 4915.5 | 69.6 | 0.93 | 0.834 | 0.152 | 0.07 | 0.166 | 0.8483 | 0.8779 | 0.75 | 0.371 | 0.667 | 4 |
| 2009 | 222.2 | 5275.4 | 77.72 | 0.907 | 0.779 | 0.202 | 0.093 | 0.221 | 0.7977 | 0.8436 | 0.69 | 0.385 | 0.641 | 5 |
| 2010 | 232.9 | 5642.5 | 87.29 | 0.883 | 0.723 | 0.262 | 0.117 | 0.277 | 0.7381 | 0.8104 | 0.64 | 0.404 | 0.619 | 6 |
| 2011 | 245.5 | 6055.8 | 91.21 | 0.855 | 0.66 | 0.286 | 0.145 | 0.34 | 0.7137 | 0.7747 | 0.59 | 0.412 | 0.594 | 7 |
| 2012 | 258 | 6469.5 | 95.13 | 0.826 | 0.596 | 0.311 | 0.174 | 0.404 | 0.6892 | 0.742 | 0.55 | 0.42 | 0.572 | 8 |
| 2013 | 270.6 | 6883.1 | 99.06 | 0.798 | 0.533 | 0.335 | 0.202 | 0.467 | 0.6648 | 0.712 | 0.52 | 0.429 | 0.553 | 15 |
| 2014 | 283.2 | 7296.8 | 103 | 0.769 | 0.469 | 0.36 | 0.231 | 0.531 | 0.6404 | 0.6843 | 0.49 | 0.438 | 0.536 | 21 |
| 2015 | 295.7 | 7710.5 | 106.9 | 0.741 | 0.406 | 0.384 | 0.259 | 0.594 | 0.6159 | 0.6587 | 0.46 | 0.448 | 0.521 | 25 |
| 2016 | 299.5 | 7639 | 106.8 | 0.732 | 0.417 | 0.384 | 0.268 | 0.583 | 0.6165 | 0.6514 | 0.46 | 0.448 | 0.52 | 26 |
| 2017 | 303.3 | 7567.6 | 106.7 | 0.724 | 0.428 | 0.383 | 0.276 | 0.572 | 0.617 | 0.6441 | 0.47 | 0.448 | 0.519 | 27 |
| 2018 | 307.2 | 7496.2 | 106.7 | 0.715 | 0.439 | 0.382 | 0.285 | 0.561 | 0.6175 | 0.637 | 0.47 | 0.447 | 0.519 | 28 |
| 2019 | 311 | 7424.8 | 106.6 | 0.706 | 0.45 | 0.382 | 0.294 | 0.55 | 0.618 | 0.63 | 0.48 | 0.447 | 0.518 | 29 |
| 2020 | 314.6 | 7353.4 | 106.5 | 0.698 | 0.461 | 0.381 | 0.302 | 0.539 | 0.6185 | 0.6237 | 0.48 | 0.447 | 0.517 | 30 |
| 2021 | 325.6 | 7610.4 | 111.4 | 0.673 | 0.421 | 0.412 | 0.327 | 0.579 | 0.5878 | 0.6049 | 0.46 | 0.46 | 0.509 | 32 |
| 2022 | 336.6 | 7867.4 | 116.3 | 0.648 | 0.382 | 0.443 | 0.352 | 0.618 | 0.5571 | 0.5871 | 0.45 | 0.473 | 0.502 | 34 |
| 2023 | 347.7 | 8124.5 | 121.3 | 0.623 | 0.342 | 0.474 | 0.377 | 0.658 | 0.5264 | 0.5704 | 0.43 | 0.487 | 0.496 | 36 |
| 2024 | 358.7 | 8381.5 | 126.2 | 0.598 | 0.303 | 0.504 | 0.402 | 0.697 | 0.4957 | 0.5546 | 0.42 | 0.502 | 0.491 | 38 |
| 2025 | 369.7 | 8638.6 | 131.2 | 0.574 | 0.264 | 0.535 | 0.426 | 0.736 | 0.4649 | 0.5397 | 0.4 | 0.518 | 0.487 | 40 |
| 2026 | 383.2 | 8865.9 | 136.8 | 0.543 | 0.229 | 0.57 | 0.457 | 0.771 | 0.4298 | 0.5225 | 0.39 | 0.538 | 0.485 | 42 |
| 2027 | 396.7 | 9093.3 | 142.4 | 0.513 | 0.194 | 0.605 | 0.487 | 0.806 | 0.3947 | 0.5064 | 0.38 | 0.559 | 0.483 | 44 |
| 2028 | 410.1 | 9320.7 | 148.1 | 0.482 | 0.159 | 0.64 | 0.518 | 0.841 | 0.3595 | 0.4913 | 0.37 | 0.582 | 0.482 | 46 |
| 2029 | 423.6 | 9548 | 153.7 | 0.452 | 0.124 | 0.676 | 0.548 | 0.876 | 0.3244 | 0.477 | 0.36 | 0.606 | 0.482 | 45 |
| 2030 | 437 | 9775.4 | 159.3 | 0.421 | 0.089 | 0.711 | 0.579 | 0.911 | 0.2893 | 0.4635 | 0.35 | 0.633 | 0.484 | 43 |
| 2031 | 449 | 9891.8 | 164.1 | 0.394 | 0.071 | 0.74 | 0.606 | 0.929 | 0.26 | 0.4522 | 0.35 | 0.658 | 0.487 | 41 |
| 2032 | 461 | 10008 | 168.8 | 0.367 | 0.054 | 0.769 | 0.633 | 0.946 | 0.2306 | 0.4413 | 0.35 | 0.684 | 0.49 | 39 |
| 2033 | 473 | 10125 | 173.5 | 0.34 | 0.036 | 0.799 | 0.66 | 0.964 | 0.2013 | 0.431 | 0.34 | 0.713 | 0.495 | 37 |
| 2034 | 485 | 10241 | 178.2 | 0.313 | 0.018 | 0.828 | 0.687 | 0.982 | 0.1719 | 0.4212 | 0.34 | 0.744 | 0.501 | 35 |
| 2035 | 497 | 10357 | 182.9 | 0.286 | 0 | 0.857 | 0.714 | 1 | 0.1425 | 0.4117 | 0.33 | 0.778 | 0.508 | 33 |
| 2036 | 507.2 | 10243 | 186.4 | 0.263 | 0.018 | 0.879 | 0.737 | 0.982 | 0.1207 | 0.4041 | 0.34 | 0.806 | 0.516 | 31 |
| 2037 | 517.3 | 10128 | 189.9 | 0.24 | 0.035 | 0.901 | 0.76 | 0.965 | 0.0989 | 0.3968 | 0.34 | 0.835 | 0.524 | 24 |
| 2038 | 527.4 | 10013 | 193.4 | 0.217 | 0.053 | 0.923 | 0.783 | 0.947 | 0.0771 | 0.3897 | 0.35 | 0.866 | 0.534 | 23 |
| 2039 | 537.6 | 9898.6 | 196.9 | 0.194 | 0.07 | 0.945 | 0.806 | 0.93 | 0.0553 | 0.3828 | 0.35 | 0.9 | 0.544 | 18 |
| 2040 | 547.7 | 9783.9 | 194 | 0.171 | 0.088 | 0.926 | 0.829 | 0.912 | 0.0737 | 0.3762 | 0.35 | 0.872 | 0.534 | 22 |
| 2041 | 555.9 | 9791.2 | 195.7 | 0.152 | 0.087 | 0.937 | 0.848 | 0.913 | 0.0632 | 0.371 | 0.35 | 0.888 | 0.538 | 20 |
| 2042 | 564.1 | 9798.5 | 197.3 | 0.134 | 0.086 | 0.947 | 0.866 | 0.914 | 0.0527 | 0.366 | 0.35 | 0.905 | 0.541 | 19 |
| 2043 | 572.4 | 9805.8 | 199 | 0.115 | 0.085 | 0.958 | 0.885 | 0.915 | 0.0421 | 0.3611 | 0.35 | 0.922 | 0.546 | 17 |
| 2044 | 580.6 | 9813.1 | 200.7 | 0.097 | 0.083 | 0.968 | 0.903 | 0.917 | 0.0316 | 0.3563 | 0.35 | 0.941 | 0.55 | 16 |
| 2045 | 588.7 | 9820.4 | 202.4 | 0.078 | 0.082 | 0.979 | 0.922 | 0.918 | 0.0211 | 0.3517 | 0.35 | 0.96 | 0.555 | 14 |
| 2046 | 595.6 | 9763.3 | 203.1 | 0.063 | 0.091 | 0.983 | 0.937 | 0.909 | 0.0169 | 0.3478 | 0.35 | 0.967 | 0.557 | 13 |
| 2047 | 602.6 | 9706.2 | 203.8 | 0.047 | 0.1 | 0.987 | 0.953 | 0.9 | 0.0127 | 0.3441 | 0.36 | 0.975 | 0.559 | 12 |
| 2048 | 609.5 | 9649.1 | 204.4 | 0.031 | 0.109 | 0.992 | 0.969 | 0.891 | 0.0084 | 0.3404 | 0.36 | 0.983 | 0.561 | 11 |
| 2049 | 616.4 | 9592.1 | 205.1 | 0.016 | 0.117 | 0.996 | 0.984 | 0.883 | 0.0042 | 0.3368 | 0.36 | 0.992 | 0.563 | 10 |
| 2050 | 623.3 | 9535 | 205.8 | 0 | 0.126 | 1 | 1 | 0.874 | 0 | 0.3333 | 0.36 | 1 | 0.566 | 9 |

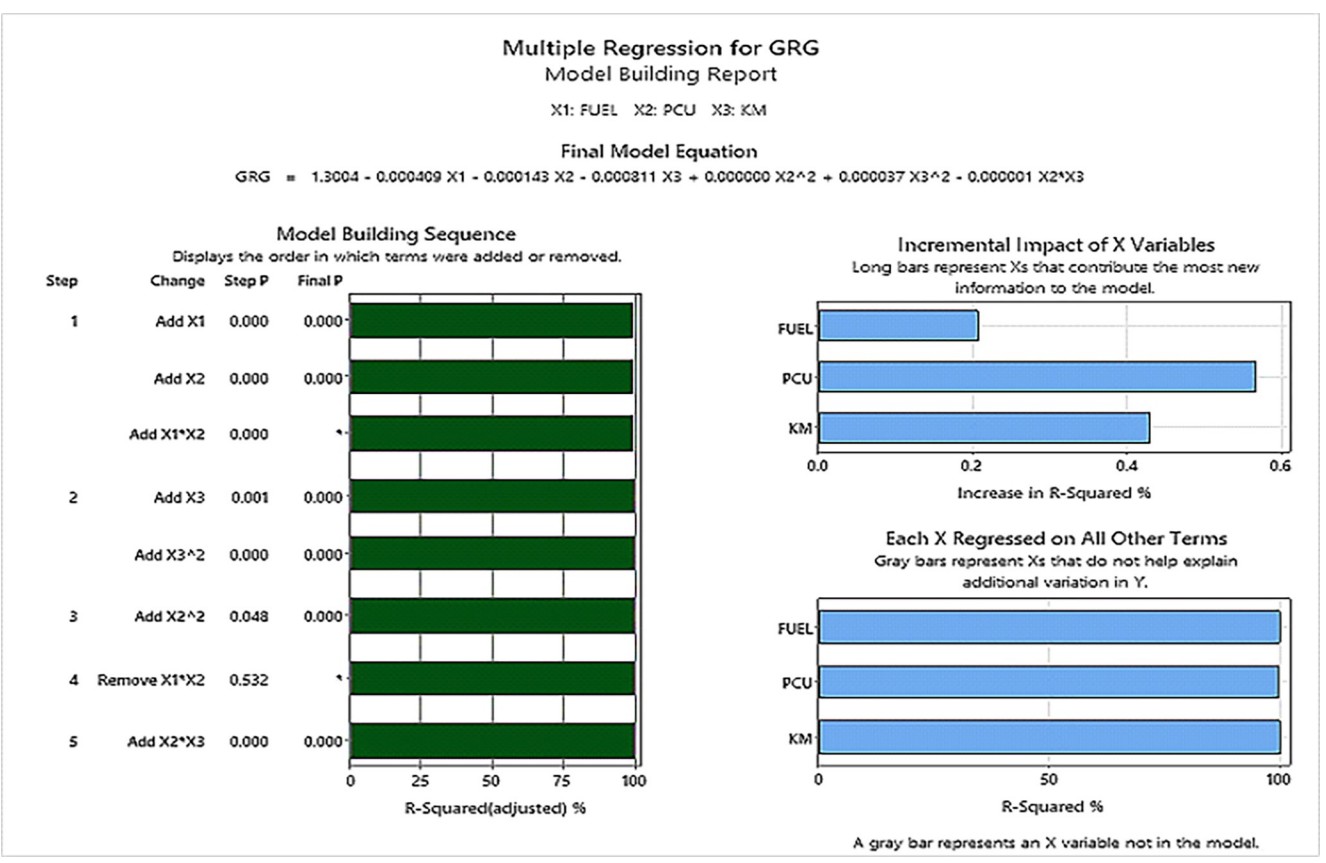

**Fig 7. Grey relational analysis for three input variables.**

Analyzing the Multi regression model for GRG, the three input variables on Minitab with p <0.01 and $R^2$ = 99.93%, the Model built is as below in Fig 7 and in S1 Fig.

$$GRG = 1.3004 - 0.000409X_1 - 0.000143X_2 - 0.000811X_3 + 0.000000X_2^2 + 0.000037X_3^2 - 0.000001X_2 * X_3.$$

The time series projection and behavior of the input and output are also seen below in Fig 8 with a steady rise in Fuel Consumption, distance traveled by the vehicle, and Grey relational grade for carbon emission. Passenger Car Units however seem to fluctuate after periodic intervals but overall the trend shows a rising pattern.

PCUs being the most significant contributor as seen in Figs 7 and 8, are further broken down into the 12 individual modal PCUs go through again the GRG Analysis for destructive PCUs. The same is also analyzed using different machine-learning techniques:

a. Random forest is a supervised machine learning and data mining technique that works on the building block of multiple decision trees where each node represents an input and each branch represents an outcome. Random Forest Regressor is a meta estimator that is applied to fit in multiple classifying decision trees on subsamples of the modes and utilizes an averaging technique to control overfitting of the data that can generate a predictive model with improved accuracy. It is a Machine Learning Algorithm applied for classification and regression function analysis to confirm the contribution of the most significant input variable. We applied Random Forest Regressor on the 12-modal road

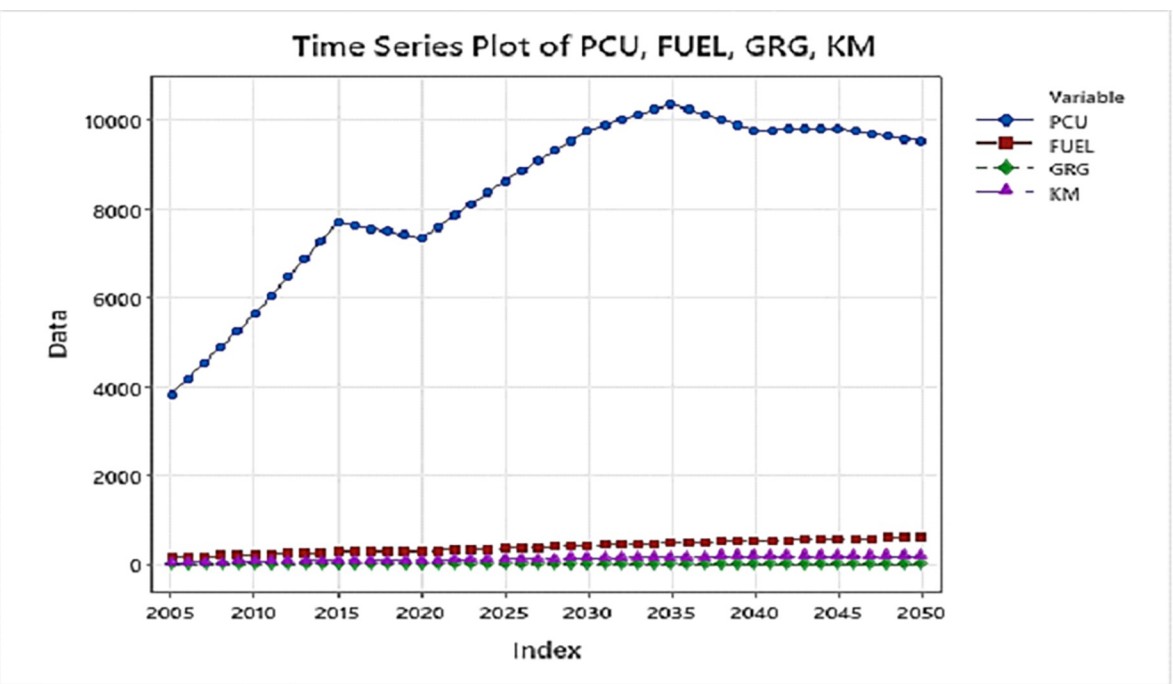

**Fig 8. Time series plot of input and output variable projection 2005–2050.** (X-axis = Index years 2005–2050; y-axis: i. Passenger Car Unit (PCU) in numbers, ii. Fuel in PJ, iii. Distance traveled in $10^9$ km & iv. GRG is a constant).

transport mix to ascertain the impact of the most significant contributor on the output. In this study, we used it to identify the most significant modal contributor to Greenhouse Gas Emissions from among the 12 modes and the analysis identified G-LD (24.516), D-HD (16.294), G-HD (15.15), D-LD (13.692), and G-C (12.0559) as the top order contributing modes with a mean absolute error of 1.17748 and mean squared error of 4.75 in the prediction model. Data is trained on a historic data set and then the model developed shows a spread of variation along data sets shown on the x-axis with $CO_2$ values on the y-axis as seen below in Fig 9(A)

b. A Decision Tree is another supervised machine learning algorithm that works both for classification and regression when applied to this problem generated the results as shown in Fig 9(B) where the predicted outcome has a mean squared error of 0.4082 and mean absolute error of 0.5840.

c. The linear Regression Model is another example of a supervised machine learning model that finds the best line fit for the dependent and independent models. This study generated the model below with a mean squared error of 1.1012e-28 and a mean absolute error of 9.76e-15 as in Fig 9(C).

The vehicle numbers in PCUs as per GAINs shows continuous growth for most of the vehicle type in this study from all the twelve modes of transport as per the GAINs Model data is expected to grow as seen in Fig 10 below:

Based on the identification of the most contributing components of the PCUs from Radom Forest the optimization study of the transport modal mix is carried out with the Grey Relational optimization technique where the top contributing GHG Emission modes are optimized based on lower the better while all other modes contributing less to GRG are

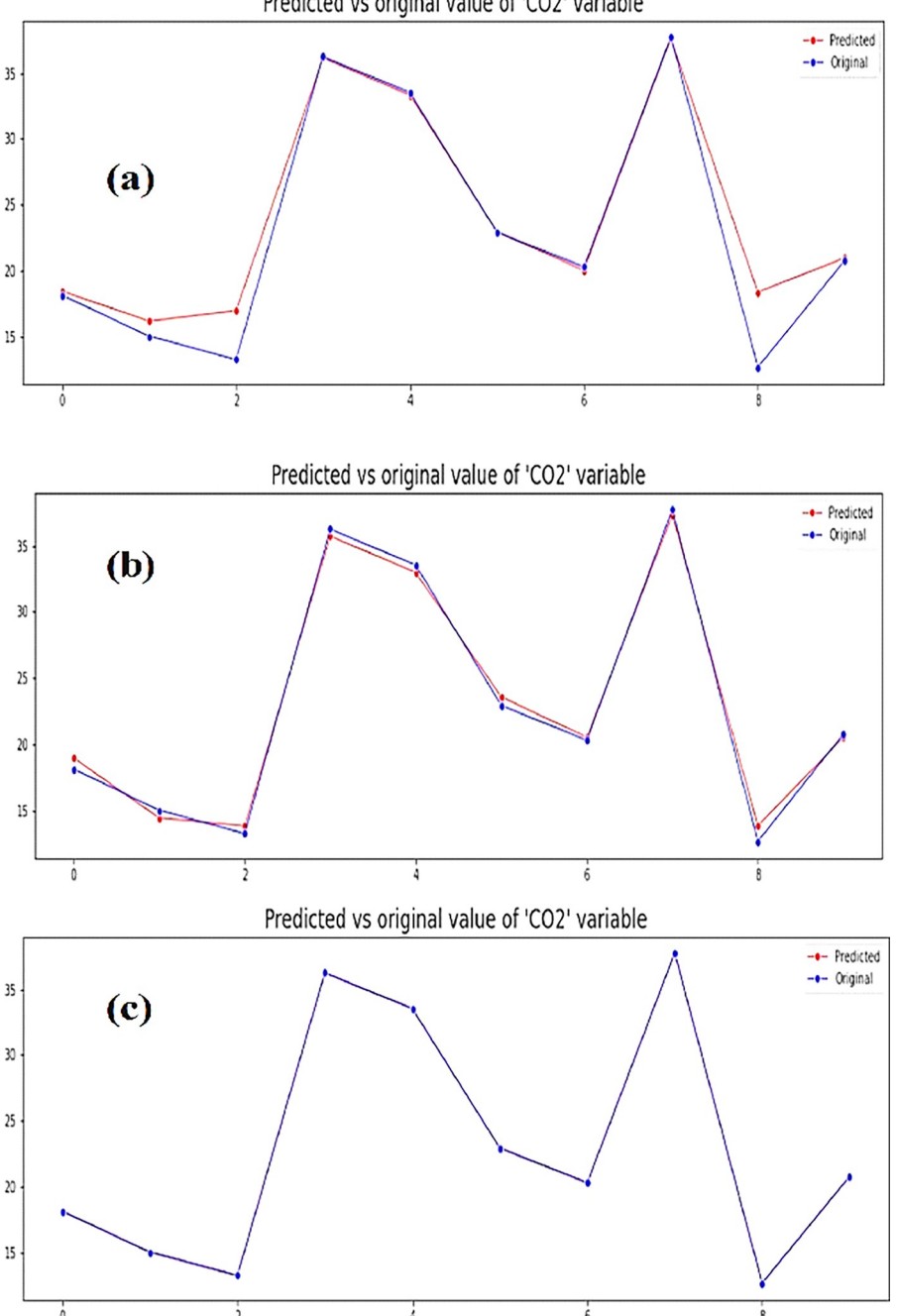

**Fig 9.** **(a)** Random Forest Prediction Model with 12 modal PCUs **(b)** Decision Tree Model **(c)** Linear Regression Model of predicted vs original values of emissions and multiple data sets.

normalized based on higher the better as shown in S3 Table. The optimized GRG output shows the year 2015 as the highest-ranking year of $CO_2$ and 2047 as the lowest ranking of emissions with PCU based on the modal mix in Fig 11.

Change in the modal mix of transport with a few modes increasing and a few reducing in number playing their role in GRG reduction from the year 2015 to the year 2050. The goal of

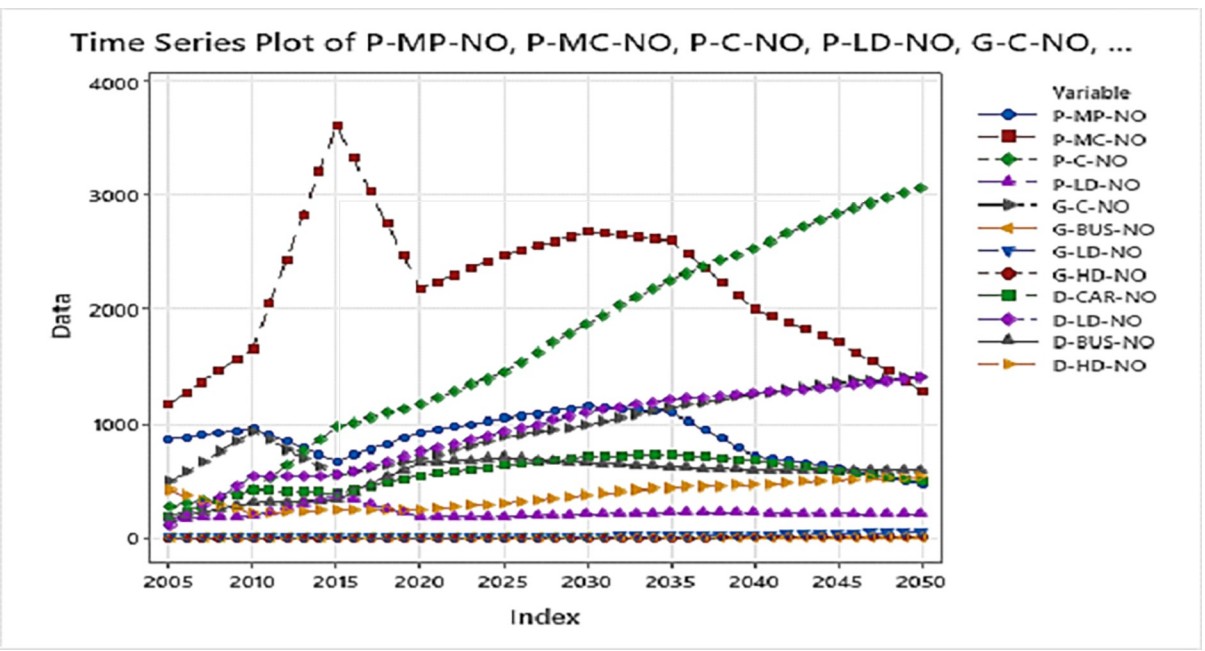

**Fig 10. Time series plot for the 12 modal inputs showing projection over 45 years from 2005–2050 while the y-axis shows the number of PCUs.**

the research is to develop the most simplistic model for the prediction of CO2 emissions therefore Artificial intelligence (AI) algorithms are sought with machine learning algorithms to generate perceptron using an activation function of relu. Different neural networks in deep learning can be used like Artificial Neural Networks (ANN), Convolution Neural Networks (CNN), and Recurrent Neural Networks (RNN). The number of layers is dependent on the complexity of the task; in our problem, the initial data analyzed 36 input parameters that are later trimmed down to 12 inputs therefore the number of layers is generated accordingly for data to learn in its forward path and compute loss function on its backpropagation. The activation function is applied to generate nonlinearity of the data and here relu is used as it is the simplest to learn from our model that works on regression and has to generate only one output parameter. In the last hidden layer, the only linear activation function is used so that it can generate straight away what it has learned from the model.

To generate better visualization of future projections of $CO_2$ emissions, a model is built in Deep Neural Network based on the 12 modal PCUs as input predicting $CO_2$ emissions as output. The mean squared error is 0.2444 while the mean absolute error from the neural network is 0.333 built on the data set from 2015–2050 with 1000/1000 epoch having 3 fully connected dense layers with 128 hidden units in the first layer, 64 hidden units in the second layer and a final layer with 1 hidden unit to determine the final $CO_2$ variable. The architecture consists of three layers with the first layer that receives the input shape in the form of (rows, and columns) as input along with a relu activation function that performs the nonlinear matrix multiplications. The second layer is also stacked on top of the first layer further performs feature extraction and nonlinear relu activation to learn complex features and the final layer with 1 hidden unit has a linear activation function that outputs the value of $CO_2$. The total parameters of the model are shown below: Model: "sequential"

| Layer (Type) | dense (Dense) | dense_1 (Dense) | Dense_2 (Dense) |
|---|---|---|---|
| Output Shape | (None, 128) | (None, 64) | (None, 1) |
| Param# | 1664 | 8256 | 65 |

**Total params:** 9,985

**Trainable params:** 9,985

**Non-trainable params:** 0

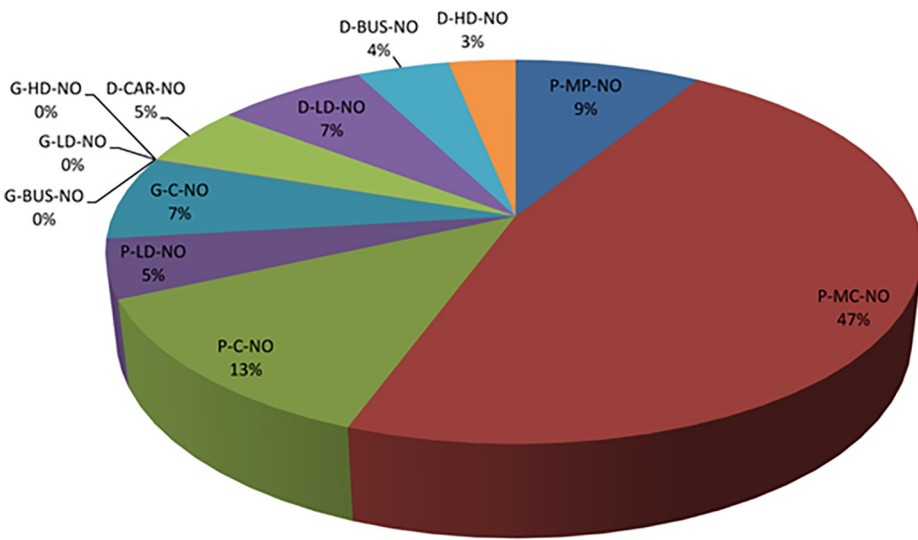

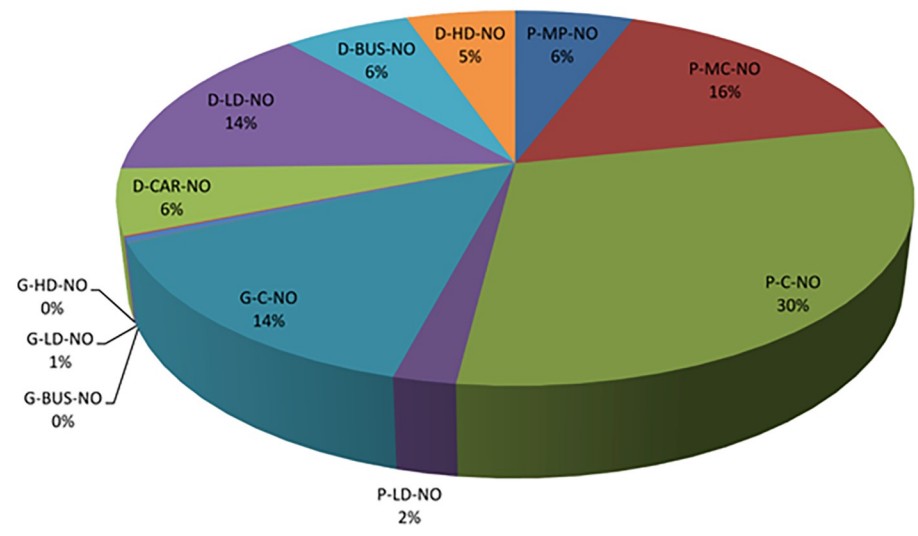

**Fig 11. The pie chart shows a modal mix % age for the years 2015 & 2047.**

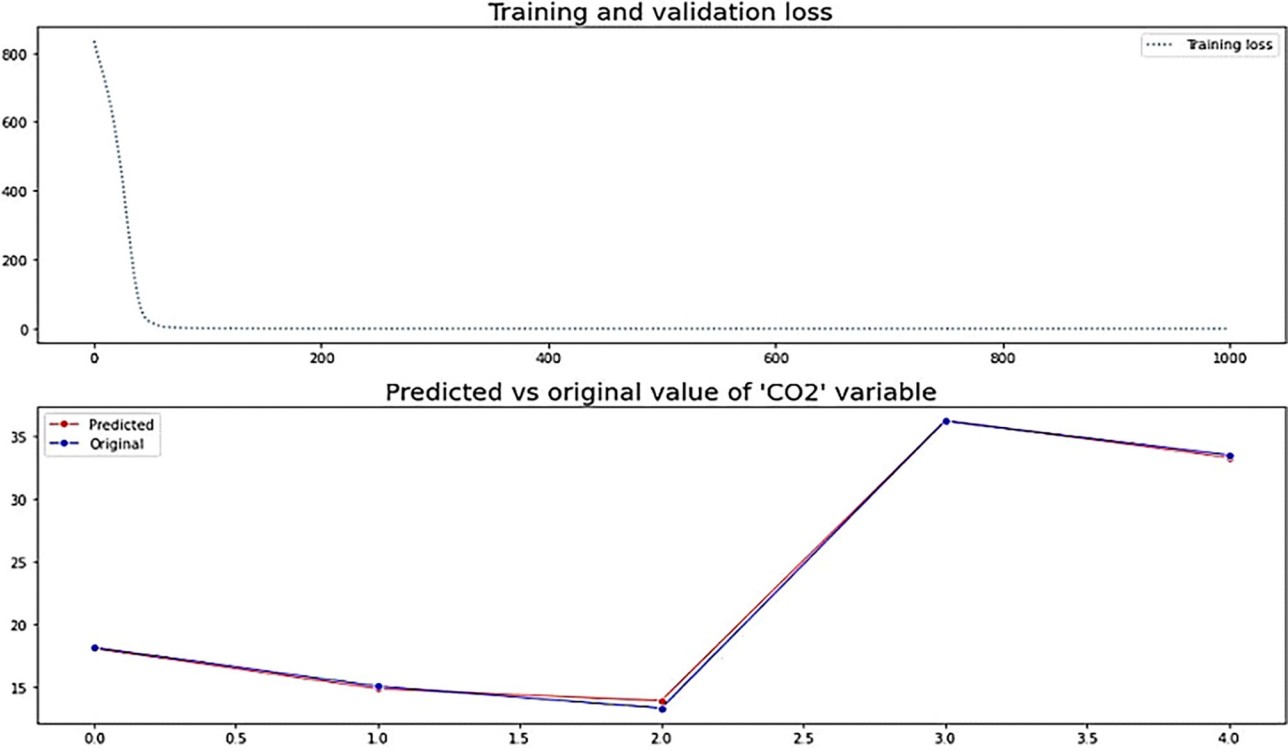

**Fig 12. Deep neural network "predicted vs original model for $CO_2$ emission" in MT.**

Overall, the model has lesser learnable parameters since we are dealing with numeric feature extraction and thus robustly fits the problem that we are solving with the difference in predicted to original values shown in **Fig 12.**

The study supports the generation of ceilings on transport mix to reduce emissions to desired levels, as promised by global INDC pledges assigned to each country. When we applied some ceilings to the model the impact on Greenhouse Gas reduction can be seen in Table 3.

The business-as-usual projection of GHG emission shows $CO_2$ emissions of 36.24 MT in the year 2044 (S3A and S5 Figs). For the baseline reading of 2044 based on business-as-usual Table 3 above suggests multiple modal mix scenarios for reduction like a 25% reduction in the number of all top emitters G-C, G-LD, G-HD, D-LD & D-HD (Mix 3) by the year 2044 suggests an overall 19.7% reduction of GHG emissions, 50% reduction suggests 26.2% reduction while the projected number of increases for these modes in the year 2044 with the year 2022 as the base year is 74.12, 1541.66, 4520, 60.24 and 84.10% respectively (S1 Table). Another scenario of Mix 1 suggests that if only a 25% reduction is achieved in the light-duty transport

**Table 3. Ceiling calculator for multimodal mix scenario generation.**

| THE YEAR 2044 | P-MP | P-MC | P-C | P-LD | G-C | G-LD | G-B | G-HD | D-C | D-LD | D-B | D-HD | $CO_2$ = 36.24MT | %REDUCTION |
|---|---|---|---|---|---|---|---|---|---|---|---|---|---|---|
| 25% | | | | | -25% | -25% | | -25% | | -25% | | -25% | 16.54 | 54.35% |
| 50% | | | | | -50% | -50% | | -50% | | -50% | | -50% | 10.04 | 72.29% |
| Mix1 | | | | +50% | | -25% | | | | -25% | | | 18.94 | 52.26% |
| Mix 2 | | | +50% | | -50% | | | -25% | | | | -25% | 25.87 | 71.38% |
| Mix 3 | | | +25% | +50% | -25% | -25% | | -25% | | -25% | | -25% | 21.88 | 60.37% |

operational by Gas and Diesel and a load of this shift falls on Petrol Light duty by adding more of this mode, even then the percentage reduction achieved is 17.3% Likewise in Mix 2 & 3 scenarios, the load is shared by Petrol Cars or Petrol Light Duty, 10.37% and 14.36% reductions can be achieved.

## 5. Conclusion

This research aims to establish quantitative emission reduction targets for the transport sector so that it can contribute to the nationally determined goals assigned to a particular country. There are only 10% of UNFCC signatory countries who have presented their transport sector emission goals in their mitigation plans, therefore, alternate emission reduction pathways and techniques are required to make the abatement plans attainable. The combination of optimization and AI climate change prediction module offers a practical solution to monitor and restrict pollutants under specified limits. The novelty of combining the techniques is its ability to break down the input variables into component-level predictor variables that can serve as a predictor of emissions. This explains the confirmation of the first research question while the second research question refers to the integration of the Optimization technique with AI tools the research confirms that AI compliments the minimax optimization of Grey relational Grade analysis and the model generated through deep neural network serves as a better emission projection sequence. The practical application of this technique is not only to generate projections for the nationally determined contribution of any country based on business as a usual basis but to develop modal mix tradeoffs.

It is learned that the incremental impact of PCUs in terms of R-squared is most significant as compared to other independent variables like fuel consumption and distance traveled. The same was further verified using a deep neural network model predicting $CO_2$ as output, however, the model has fewer learnable parameters, and the output in the numeric figure is based on the difference in predicted to the actual values. The minimax function identifies the most damaging modes and proposes a method based on backward integration with intermodal switching choices to achieve reduced carbon emissions years ahead. The highest contributors to transport emissions are Heavy-duty vehicles and buses whether using diesel or gas along with Petrol light-duty vehicles due to their vehicle number while the least contributors are Petrol Moped and Motorcycles. The number of vehicles and their combination is termed a transport mix and when used in backward integration can generate emission ceilings.

It can therefore be concluded that a simplified multimodal transport mix model promises to give a quick and more efficient emission reduction target using this combination of minmax and AI techniques on region-specific data. The restriction for different regions based on their business-as-usual projection can be transformed as policy implications by applying ceilings on specific transport modes functioning in that region. In our current study of the Punjab Province of Pakistan, after training the model, when ceilings were assigned to the top polluting modes of transport, we can see the impact on emissions from the year 2022 to the year 2044. It is observed that by reducing HD vehicles by 25% from both diesel and gas, a 19.7% reduction in carbon emissions is predicted. Likewise, a 50% reduction in HD diesel and gas vehicles would achieve a 26.2% reduction in emissions. This reduction coupled with a tradeoff scenario would generate more rational results like a 25% reduction in both Diesel and Gas Light-duty vehicles respectively would lead to a 50% increase in Petrol Light Duty vehicles resulting in a 17.3% reduction in carbon emissions.

The theoretical contribution of the study is an efficient first-hand tool to generate multiple scenarios of the modal mix by suggesting different combination strategies for the transport mix in a country. The outcome can be supported further by traffic policy implications for the

least emission releases. The HD freight vehicles likewise can be shifted to railways or waterways depending on the availability of accessible alternates. This can help bring a significant reduction in emissions by putting a check on vehicle numbers of varied modes and calculating tradeoffs of emissions involved in the modal shift from land to rail and air from the emission perspective. Modification measures like staggered office hours, pedestrian flow, vehicle numbers on the road during rush hours in urban areas, and alternate freight modes. There are certain limitations to this technique as the technique is not generic but customized and the model becomes specific for every country or region suggesting ceilings on the number of vehicles of each mode specific to the dynamics of that region. However, to make the INDC goals achievable every country has to establish its policy that can cater to the dire requirement for intermodal transport integration between land, sea, and air modes. Even in the road transport category, the preferred vehicle mode can be selected to optimize the usage of low-emission modes. An intermodal shift has its tradeoffs and each change would bring an impact on the vehicle number, fuel consumed, and emissions released. The future application of this study is to apply the technique on intra-modal tradeoffs in transport from the emission perspective and policy implication perspectives for a cleaner environment in the future.

## Supporting information

**S1 Fig.**
(TIF)

**S2 Fig.**
(TIF)

**S3 Fig.**
(TIF)

**S4 Fig.**
(TIF)

**S5 Fig.**
(TIF)

**S1 Table.**
(TIF)

**S2 Table.**
(TIF)

**S3 Table.**
(TIF)

## Author Contributions

**Conceptualization:** Huma Rauf.

**Data curation:** Huma Rauf.

**Formal analysis:** Huma Rauf.

**Investigation:** Huma Rauf.

**Methodology:** Huma Rauf.

**Project administration:** Huma Rauf.

**Resources:** Huma Rauf.

**Software:** Huma Rauf.

**Supervision:** Muhammad Umer.

**Validation:** Huma Rauf, Muhammad Umer.

**Visualization:** Huma Rauf.

**Writing – original draft:** Huma Rauf.

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
