## [Decision Letter · Decision Letter 0]

18 Apr 2023

PONE-D-23-04895Development of AI-Augmented Optimization Technique for Analysis & Prediction of Modal Mix in Road TransportationPLOS ONE

Dear Dr. Rauf,

Thank you for submitting your manuscript to PLOS ONE. After careful consideration, we feel that it has merit but does not fully meet PLOS ONE’s publication criteria as it currently stands. Therefore, we invite you to submit a revised version of the manuscript that addresses the points raised during the review process.

We look forward to receiving your revised manuscript.

Kind regards,

Muhammad Kamran Khan, PhD Finance

Academic Editor

PLOS ONE

Reviewers' comments:

Reviewer's Responses to Questions

**Comments to the Author**

1. Is the manuscript technically sound, and do the data support the conclusions?

Reviewer #1: Partly

2. Has the statistical analysis been performed appropriately and rigorously? 

Reviewer #1: Yes

3. Have the authors made all data underlying the findings in their manuscript fully available?

Reviewer #1: Yes

4. Is the manuscript presented in an intelligible fashion and written in standard English?

Reviewer #1: Yes

5. Review Comments to the Author

Reviewer #1: Dear authors,

I'd like to congratulate you and your team on your excellent research work in your paper submitted for publication in this prestigious journal. The topic is very interesting, and I enjoyed it. I would like to thank you for your efforts in presenting your research work in such a professional manner. However, before your work is recommended or accepted, a few comments must be included/ addressed to improve the quality of your work as well as for future publication in this reputable journal. I have the following observations, questions, and comments that may help to improve your work. The authors must modify the following points in great detail.

1. In the abstract, please include 2-3 special quantitative achievements from the findings of this study in the context of the environment by combining the research objectives and problems. Please limit your abstract to 250 words. Check spellings for many words that are misspelt or written in haste.

2. The introduction section needs a few more sentences to strengthen the article, and please include the research problem, objective, and novelty in the last paragraph of the Introduction section.

3. Include a few more sentences at the beginning of the introduction explaining your paper's contribution to the environment, climate change impact, and sustainability, as well as your attempts to deal with or present solutions to a specific problem/s and your unique contribution with this research paper.

4. Please also present the methodology section in a concise graphical format.

5. The literature review section is very weak; please revise it.

6. Please present your literature review in the form of a SmartArt chart.

7. Just after the Methodology, please mention the societal benefits of your research in terms of evaluating its key determinant.

8. In 500-750 words, explain research problems, solutions, and the theoretical contribution of your study in the "Results" section.

I found that the literature section is a little weak, shift your study a little more towards environment friendly and sustainability, therefore it requires more studies to be reviewed therefore I suggest you to include the following work:

https://doi.org/10.1016/j.strueco.2022.04.003

https://doi.org/10.1016/j.envres.2022.112848

https://doi.org/10.1108/MD-03-2022-0407

https://doi.org/10.1007/s11356-022-20567-6

https://doi.org/10.1177/09763996211041215

https://doi.org/10.1007/s11356-023-25574-9

https://doi.org/10.1007/s11356-021-15421-0

I think above all studies will make this study more relevant in bridging the gap with literature.

Looking forward for your revised submission.

6. PLOS authors have the option to publish the peer review history of their article (what does this mean?). If published, this will include your full peer review and any attached files.

Reviewer #1: **Yes: **Vishal

<quillbot-extension-portal></quillbot-extension-portal>

---

## [Author Response · Author response to Decision Letter 0]

9 Jun 2023

June 1, 2023

Reference: PONE_D-23-04895, Development of AI-Augmented Optimization Technique for Analysis & Prediction of Modal Mix in Road Transportation for PLOS ONE

Dear Dr. Muhammad Kamran Khan, 

Kindly refer to the email dated April 18, 2023, regarding the revision requirement of the referenced article, I am pleased to submit the revised article with this letter, and compliance is mentioned below:

Journal Requirements: 

S.No. Requirement Compliance

Templates for both title & author affiliation along with the main body followed in the manuscript submitted. 

2. We note that you have stated that you will provide repository information for your data at acceptance. Should your manuscript be accepted for publication, we will hold it until you provide the relevant accession numbers or DOIs necessary to access your data. If you wish to make changes to your Data Availability statement, please describe these changes in your cover letter and we will update your Data Availability statement to reflect the information you provide. Data is already been made available in the paper and changes in the Data Availability statement are made accordingly. 

Reviewer #1 Comments: 

S.No. Requirement Compliance

1. In the abstract, please include 2-3 special quantitative achievements from the findings of this study in the context of the environment by combining the research objectives and problems. Please limit your abstract to 250 words. Check spellings for many words that are misspelled or written in haste. Revised and Complied

2. The introduction section needs a few more sentences to strengthen the article, and please include the research problem, objective, and novelty in the last paragraph of the Introduction section. Revised and Complied

3 Include a few more sentences at the beginning of the introduction explaining your paper's contribution to the environment, climate change impact, and sustainability, as well as your attempts to deal with or present solutions to a specific problem/s and your unique contribution to this research paper. Revised and Complied

4 Please also present the methodology section in a concise graphical format. Revised and Complied

5 The literature review section is very weak; please revise it. Revised and Complied

6 Please present your literature review in the form of a SmartArt chart. Revised and Complied

7 Just after the Methodology, please mention the societal benefits of your research in terms of evaluating its key determinant. Revised and Complied

8 In 500-750 words, explain research problems, solutions, and the theoretical contribution of your study in the "Results" section.

I found that the literature section is a little weak, shift your study a little more towards environment friendly and sustainability, therefore it requires more studies to be reviewed therefore I suggest you to include the following work:

https://doi.org/10.1016/j.strueco.2022.04.003

https://doi.org/10.1016/j.envres.2022.112848

https://doi.org/10.1108/MD-03-2022-0407

https://doi.org/10.1007/s11356-022-20567-6

https://doi.org/10.1177/09763996211041215

https://doi.org/10.1007/s11356-023-25574-9

https://doi.org/10.1007/s11356-021-15421-0

Revised and Complied. All mentioned papers are included in the paper. 

 I hope to have replied to all revision requirements and sincerely look forward to the acceptance of this paper for publication in your esteemed Journal. 

Sincerely,

Huma Rauf

Department of Business & Engineering Management, 

Sir Syed CASE Institute of Technology (SSCIT), B-17

Islamabad 44400, Pakistan

---

## [Decision Letter · Decision Letter 1]

29 Jun 2023

Development of AI-Augmented Optimization Technique for Analysis & Prediction of Modal Mix in Road Transportation

PONE-D-23-04895R1

Dear Dr. Rauf,

We’re pleased to inform you that your manuscript has been judged scientifically suitable for publication and will be formally accepted for publication once it meets all outstanding technical requirements.

Kind regards,

Muhammad Kamran Khan, PhD Finance

Academic Editor

PLOS ONE

Additional Editor Comments (optional):

Reviewers' comments:

Reviewer's Responses to Questions

**Comments to the Author**

1. If the authors have adequately addressed your comments raised in a previous round of review and you feel that this manuscript is now acceptable for publication, you may indicate that here to bypass the “Comments to the Author” section, enter your conflict of interest statement in the “Confidential to Editor” section, and submit your "Accept" recommendation.

Reviewer #1: All comments have been addressed

2. Is the manuscript technically sound, and do the data support the conclusions?

Reviewer #1: Yes

3. Has the statistical analysis been performed appropriately and rigorously? 

Reviewer #1: Yes

4. Have the authors made all data underlying the findings in their manuscript fully available?

Reviewer #1: Yes

5. Is the manuscript presented in an intelligible fashion and written in standard English?

Reviewer #1: Yes

6. Review Comments to the Author

Reviewer #1: Dear Author/s,

I would like to congratulate you and your team for doing such a good research work in your submitted paper. Topic is very interesting and I liked the topic and appreciate your efforts to present your revised research work in such a nice manner. I am satisfied from your efforts you employed in the revision and I found all my suggested comments have been incorporated or addressed perfectly. Therefore, I strongly recommended this article for acceptance for further publication in this reputed journal without any more changes.

The revised version of the paper looks perfect, in the section of the literature review the very relevant, connected, and updated with new references.

Methodology mentions this research's socio-economic benefits in evaluating its crucial determinant.

The section on results in the revised version explains all of the tables more briefly, and the explanations for each table in the "Results" sections, define more visibility of empirical results.

In the revised version after adding about 150 words to the conclusion section related to policy implications explaining and connecting the future scope of your research study, any limitations encountered while conducting your research, and the procedure for removing research limitations.

The revised version communicates the introduction which provides helpful information about the intended topic; however, it needed to be revised to make it meaningful and the authors are successfully able to revise it. Indeed, the research GAP is now well explained in a scientific way. Therefore, the research can be recommended for publication without any more modification in its the current form which provides sufficient justification and arguments.

7. PLOS authors have the option to publish the peer review history of their article (what does this mean?). If published, this will include your full peer review and any attached files.

Reviewer #1: **Yes: **vishal

---

## [Editor Report · Acceptance letter]

6 Jul 2023

PONE-D-23-04895R1 

Development of AI-Augmented Optimization Technique for Analysis & Prediction of Modal Mix in Road Transportation 

Dear Dr. Rauf:

I'm pleased to inform you that your manuscript has been deemed suitable for publication in PLOS ONE. Congratulations! Your manuscript is now with our production department. 

Kind regards, 

on behalf of

Dr. Muhammad Kamran Khan 

Academic Editor

PLOS ONE